# Safe and Sustainable Design of Composite Smart Poles for Wireless Technologies

**Donato Di Vito** [1] , **Mikko Kanerva** [2,*] , **Jan Järveläinen** [3] , **Alpo Laitinen** [4] , **Tuomas Pärnänen** [2] , **Kari Saari** [4] , **Kirsi Kukko** [4] , **Heikki Hämmäinen** [5] and **Ville Vuorinen** [4]

1 Faculty of Information Technology and Communication Sciences, Tampere University, FI-33014 Tampere, Finland; Donato.divito@tuni.fi

2 Faculty of Engineering and Natural Sciences, Tampere University, FI-33014 Tampere, Finland; Tuomas.parnanen@tuni.fi

3 Premix Oy, FI-05200 Rajamäki, Finland; Jan.jarvelainen@premixgroup.com

4 Department of Mechanical Engineering, Aalto University, FI-00076 Espoo, Finland; Alpo.laitinen@aalto.fi (A.L.); Kari.saari@aalto.fi (K.S.); Kirsi.kukko@aalto.fi (K.K.); Ville.vuorinen@aalto.fi (V.V.)

5 Department of Communications and Networking, Aalto University, FI-00076 Espoo, Finland; Heikki.hammainen@aalto.fi

\* Correspondence: Mikko.kanerva@tuni.fi; Tel.: +358-40-718-8819

**Abstract:** The multiplicity of targets of the 5G and further future technologies, set by the modern societies and industry, lacks the establishment of design methods for the highly multidisciplinary application of wireless platforms for small cells. Constraints are set by the overall energy concept, structural safety and sustainability. Various Smart poles and Light poles exist but it is challenging to define the design drivers especially for a composite load-carrying structure. In this study, the design drivers of a composite 5G smart pole are determined and the connecting design between finite element modelling (FEM), signal penetration and computational fluid dynamics (CFD) for thermal analysis are reported as an interdisciplinary process. The results emphasize the significant effects of thermal loading on the material selection. The physical architecture, including various cutouts, is manipulated by the needs of the mmW radios, structural safety and the societal preferences of sustainable city planning, i.e., heat management and aesthetic reasons. Finally, the paint thickness and paint type must be optimized due to radome-integrated radios. In the future, sustainability regulations and realized business models will define the cost-structure and the response by customers.

**Keywords:** tubular composites; finite element analysis; computational fluid dynamics; wireless communication; signal attenuation

## 1. Introduction

### 1.1. Wireless Outdoor Platforms

The application spectrum enabled by the fast 5G development is about to cover a multiplicity of wireless technologies and services. The selection of the frequencies for '5G' has been globally discussed and, in Europe, the focus is on the 3.6 GHz and 26 GHz bands. The Electronic Communications Committee 'ECC' conducted a survey already 2017 that suggested the bands of 24.25–27.5 GHz, 40.5–43.5 GHz and 66–76 GHz as the prioritized bands. The higher end of the radio frequency (RF) band range directly affects the radio configuration and energy usage. The needs for higher data rates and the available RF bands have led to the concepts of small cell networks, the necessary low latency, new business environment with the end to end networks [1–3] and the emphasis on cost

distribution. Besides, new terms, such as Smart pole or Light pole, known as concepts for 5G-enabling poles (5GPs) and heavier 5G gantries have been exhibited. Several demo designs or even demo sites with certain pole designs are built and running—yet many of them lack 5G operation or having a partial 5G operation. In few years, the number of 5GPs proposals in public has increased from a few to number of designs. Yet, the overall 5GP concept along with data management is still to be explored.

The strategic importance of 5GPs as a platform for 5G outdoor small cells finally stems from its costs and regulations applied. Because of the large number of required 5G base station sites, the cost of deployment and operation is high even for national mobile network operators (MNOs). A simple wholesale site contract on 5GPs with the city council is lucrative compared to hand-picking and tailoring contracts for non-uniform sites on private buildings. This encourages MNOs toward 5GP sharing for the high frequency deployments.

On the other hand, the increasing pressure to unify and beautify the city antenna 'jungle' also supports the sharing of well-designed 5GPs. Therefore, the national regulators may allow extending the monopolies of light pole and grid networks with another monopoly, a neutral host company operating the 5GP pole system [4]. According to initial studies, the cost of the structural part (the pole shaft) is significant—between 15–25% of the total deployment cost. This cost share varies depending on the amount of electronics of various services per pole and on the density of fully configured 5G smart light poles [5]. Interestingly, the faster price erosion of electronics seems to gradually increase the relative value of the poles shafts. The 5G data pricing models tend to be even more complex since it is currently not clear who sells and what kind of products.

Clearly, the modularity of devices plays a role in the prospective design of services per 5GP site. The modularity is also a tool to handle the cost structure per type of site and at individual 5GPs. The integration of a 5GP requires connections to data, power and possibly cooling network of the city or suburb. The development of design tools combining the wireless network and city planning is essential and is a significant design phase in the future. A sophisticated smart pole, a 5GP, can be considered as aesthetically fitted integrated structure, which embodies various devices including the radios within the main structure of the pole.

*1.2. Design Drivers for a Smart Pole Structure*

The general, main design drivers, are illustrated in Figure 1. The physical frame, referred to as pole or shaft, is needed primarily to carry the (RF) radios and other electrical devices in a functional and maintenance-friendly way. Since the transmitting and receiving electronics require protection against weather—irrespective if they are fully integrated inside the pole or not—the selected structural materials must possess known, specified interference with RF signals to allow for a dense pole population operating over the specified frequency range. Especially, when customers, such as cities, require for an operational lifetime of 20...50 years for each pole, durability is essential as well.

For many countries, the safety in terms of vehicle crash must be accounted for in the design of the 5G pole's shaft. The crash design affects primarily the pole sites with a high traffic density. Thus, an electrical vehicle (EV) charging stations or similar services mounted low are not allowed for these sites. The safety and overall sustainable operation of 5G is essential in general [6], since the citizens and their considerations justify the realized 5GPs at urban areas. The sustainability of the manufacture and minimum usage of material in 5GPs depends clearly on the realized, future operation time. Whenever material can be recycled, the selection will affect the sustainability in the big picture of future operation [7]. In an even wider perspective, the sustainability of an individual pole and manufacture covers only part of the truth. The thermal management and energy efficiency of the devices attached to the pole have a cardinal role. For certain pole sites, centralized cooling might form an important effect on the over-lifetime carbon and energy foot print of the entire wireless platform.

For modern, dense-packed electronics [8], thermal properties and heat management are an essential part of the system design to prevent overheating during the anticipated operation. Fluid dynamics and heat transfer are the key fields of science regarding thermal management.

The state-of-the-art numerical approach for simulating heat transfer with solid-liquid or solid-air interfaces is computational fluid dynamics (CFD). Using CFD simulations, thermal assessment on 5GPs can also be carried out. In a previous work, air cooling of high-power electronics was investigated inside a tubular pole-like structure [9]. However, the flow control can be challenging in complex environments, such as inside a tight-packed smart pole shaft.

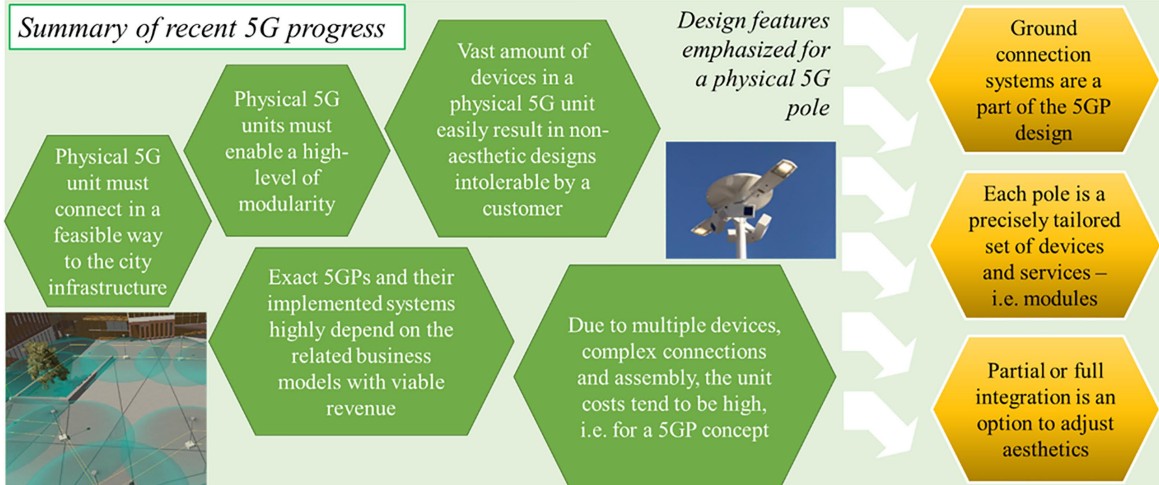

**Figure 1.** Summary of the design drivers for a physical platform for various wireless devices.

### 1.3. The Advantages and Sustainability of Composite Materials

The advantages of composite materials in terms of mechanical properties and weight are well-known. Similarly, in a smart pole, the mechanical properties, i.e., high stiffness to prevent large sway due to the wind and sun as well as the low weight for sustainable transportation and overhaul are advantages. The overall advantage of composite materials is the ability to be precisely tailored [10]. Composites have been successfully used for various shaft structures with fully composite or hybrid material lay-up [11,12]. The raw material costs and manufacture can be affected by the selection of proper fibre and matrix. Polyester resins as well as polypropylene have been applied in composites as the matrix component [13,14] in order to have a suitable balance between costs and performance. The smart pole shafts could also be made of natural fibre reinforced composites [15,16]. However, the use of natural materials requires clearly more information about sustainable fillers for fire retardancy [17] and also about the susceptibility of natural fibres to moisture [18] in long term outdoor operation.

In general, glass fiber reinforced polymers have been applied in challenging applications, such in wind turbine blades [19] where the composite reaches the extreme requirements of fatigue life and durability. Although polyesters represent the 'low' performance of composites, their benefit is the well-known behaviour in various environments [20,21] and lower material costs. From the point of view of enclosure functionality, fibrous composites can be hybridized with metal sheets to control the electromagnetic response [22]. With fiber-metal hybrids, enclosures can be made to totally protect sensitive electronics against external, harmful or unwanted radiation and signals [23]. As an alternative, particle inclusions [24] can be used to control the signal penetration.

Whenever signal penetration is necessary through the enclosure or shaft wall, the material selection becomes challenging. Polymers, typically used as matrix in composites, incur low electromagnetic attenuation in terms of dielectric loss—especially for frequencies below the gigahertz-regime. For the higher frequencies, already the type and grade of the polymer blend must be well optimized [25,26]. Reinforcing fibers are generally not especially advantageous in terms of signal penetration—carbon fibers and all conductive fibers lead to very high attenuation. Even when using glass or polymeric (e.g., aramid) fibers, the more or less sporadic orientation at a micro-scale and their multi-interface configuration within the matrix result in reflections and attenuation. Moreover, any

accumulated moisture has significant effects on attenuation. Anyhow, a finite amount of attenuation can be accepted while zero penetration for metal alloys is not an option. The tailoring of composites refers to, along with the mechanical and electromagnetic properties, thermal properties. Requirements for the matrix and fiber selection can also include limits for the thermal conductance and expansion. It should be emphasized that the thermal expansion in a composite material can be tailored by the selection of fibre and matrix as well as by the decision of lay-up.

In this study, we focus on a physical application for wireless technologies, i.e., an integral smart pole. The research targets to offer the physical platform and process for the latest 5G implementations with a multiplicity of functional requirements along with the structural safety. Whenever the (5G) wireless devices and services require highly precise and stable location in a pole, meaning minimum sway in various environments, the mechanical design must be united with the functional and thermal design. The main phases of the process described in this study are shown in Figure 2.

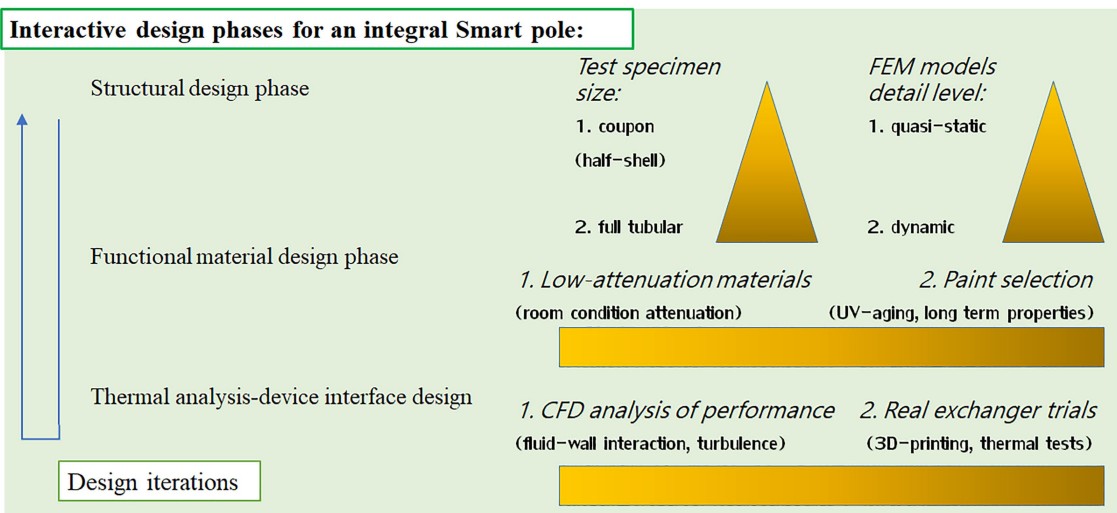

**Figure 2.** A graph of the main analysis phases for an integral smart pole allowing current 5G services.

## 2. Materials and Methods

### 2.1. Pole Structure

The pole structure for experimental tests in this study was a hollow pull-winded (specialized pultrusion) composite profile. The reinforcing fiber was a commercial E-glass fiber yarn (4800 tex, filament diameter $\approx 23$ μm, Europe) for the structural layers and ECR-glass for the surfacing mat. Glass fibre reinforced plastic (GFRP) was formed by using polyester (Norsodyne P 46074, Polynt, Italy) as the matrix constituent. The shaft was pultruded by Exel Composites Plc. (Finland) and a lay-up of [0°, 85°, 0°] was finally applied with nominal layer thicknesses of 2.8, 0.4, 2.8 mm, respectively. The shaft had a constant outer diameter of $D_{outer} = 168$ mm and as-received nominal wall thickness of $t_{wall} = 6$ mm. The surfacing mat was used due to aesthetic reasons and had its thickness comparable to the standard manufacture deviation in the pole thickness.

### 2.2. Impact Dynamics of the Shaft GFRP

Quasi-static (QS) response by indentation was measured from half-circular (180°) panels (see Figure 3a) of the GFRP profile ($D_{outer} = 168$ mm, see Section 2.1). The panel specimen (length of 250 mm) was supported by two half-circular steel sections, which were set 200 mm apart (defining the measurement area). The upper side of the specimen was supported in the areas of steel sections resulting in a semi-rigid boundary condition. The specimen was loaded in a universal testing machine (30 kN load cell, model 5967, Instron, High Wycombe, UK) by a hemispherical head (radius 10 mm). The loading was subjected in the middle of the specimen by using a test rate of 1.0 mm/min. 3D Digital

image correlation (DIC) was used to record and analyze (Davis 8.4 software, LaVision, Göttingen, Germany) the displacement on the lower surface of the GFRP panel. In addition, the contact force was measured by the load cell located over the loading head.

The testing was continued on the panel specimens loaded by a falling/drop-weight impactor (FWI Type 5, Rosand, Leominster, Herefordshire, UK) (see Figure 3b). Instead of the semi-rigid support, an open boundary was used for the half-circular specimen, i.e., the upper side of the specimen was able to deform freely during the dynamic loading. The specimen was loaded by a hemispherical impactor head (radius 10 mm) weighting 7.67 kg with an impact energy of 100 J. The contact force of the impactor head was measured by a piezo-electric load cell located above the impactor (Type 9031A, Kistler, Winterthur, Switzerland). The displacement of the impactor head was calculated analytically from the contact force-time response.

In the last test step, a tubular shaft specimen was tested using the drop-weight impactor (Figure 3c). A shaft length of 780 mm was used in a cantilever support mode and loaded with a half-circular shaped (2D) impactor head weighting 8.49 kg. The impact energy in the testing was 100 J. Similar to the testing of the half-circular specimen, the contact force of the impactor head was measured by the piezo-electric load cell and the displacement of the impactor head was calculated analytically from the contact force-time response.

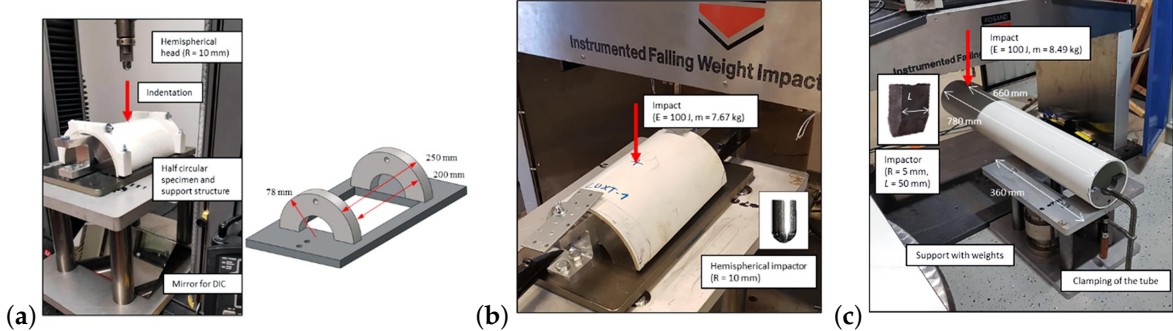

**Figure 3.** Testing of GFRP mechanics: (**a**) the QS indentation of a half-circular specimen with a semi-rigid boundary; (**b**) the drop-weight impact testing of a half-circular specimen with an 'open' boundary, and; (**c**) the drop-weight impact testing of a tubular pole specimen.

### 2.3. Finite Element Modelling and Mechanical Analysis

In this study, two different shaft cross-sections, i.e., shaft designs, were analyzed: (1) traditional circular profile and (2) a near-rectangular diamond-shaped profile. The near-rectangular cross-section, as illustrated in Figure 4, can accommodate radios, other devices and cabling while optimizing the coverage of the transmitting devices around the surroundings of the pole. Moreover, such design can be further divided into internal slots to manage the cabling and piping inside the pole and to separate different equipment by standard modules.

The composite shaft structure—the two cross-sections—were modelled by using the ABAQUS standard/explicit (2017) software code. In order to widely characterize the behaviour and loading conditions, different 3D models of the shaft were analyzed with finite element analysis (FEA). The full-scale model of the pole was simulated to characterize deformation under a mechanical load and thermal expansion due to temperature increase by solar radiation. Furthermore, material was removed in specific pole locations of the model to simulate the machining processes needed to accommodate and mount electronic devices (i.e., via so-called maintenance cutouts and doors) in the root and top section of the pole. In this case, curvature radii as well as dimensions of the holes were analyzed, together with their effect on the pole's structural integrity for the static loads mentioned above. The results obtained were then used to properly design the cutouts for electronic components and maintenance doors also at the base of the 5GP. In order to keep the consistency of the results between the different analyses, continuum shell elements (SC8R) were used in all of the finite element

models of the pole shaft. The mechanical properties of the laminate and the local impact performance have been reported elsewhere [27,28]; the elastic constants used in the modelling here are shown in Table 1 and the strength values for the laminate are 500 MPa, 200 GPa, 50 MPa, 100 MPa and 40 MPa for the longitudinal tensile strength, longitudinal compressive strength, tensile transverse strength, compressive transverse strength and shear strength, respectively.

**Table 1.** Elastic constants of the GFRP (unidirectionally reinforced layer) used in this work for the pole shaft FEA [27,28].

| Constant | Parameter | Value (Units) |
|---|---|---|
| Axial Young's modulus | $E_{11}$ | 35 GPa |
| Transverse Young's modulus | $E_{22}$ | 7.0 GPa |
| Out-of-plane Young's modulus | $E_{33}$ | 4.5 GPa |
| Poisson's ratios | $\nu_{12}, \nu_{23}$ | 0.3 |
| Poisson's ratio | $\nu_{13}$ | 0.05 |
| Shear modulus | $G_{12}$ | 3.6 GPa |
| Shear modulus | $G_{13}$ | 3.6 GPa |
| Shear modulus | $G_{23}$ | 2.0 GPa |

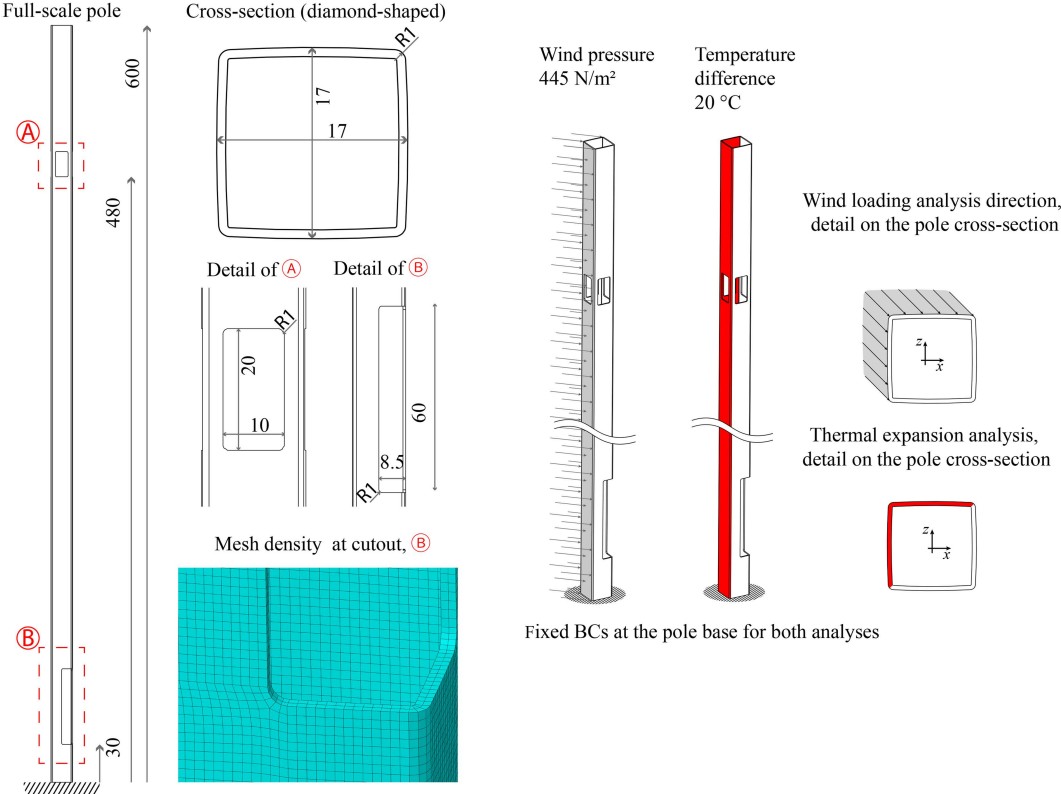

**Figure 4.** The dimensions (units: cm), loading, boundary conditions and mesh density applied to FEA.

For most of the composite materials, the thermal expansions are lower than those for any steel alloy. Thermal expansion coefficients of fibrous composites depends on the applied constituent materials and the lay-up of reinforcement, along with many factors of measurements [29,30]. In this study, the linear thermal expansion coefficients (CTEs) of the GFRP composite were determined using an experimental arrangement and an iterative FEA of the CTE values based on the residual strain, as described in Section 2.4 and Appendix A. The boundary conditions and loads (wind load distribution), shown in Figure 4, reflect a standard specification [31]. The wind loads represent the primary mechanical load against which the pole structure must be verified. The wind reference speed is given by the standard, e.g., here it is 21 m/s for the terrain category II of Scandinavia. The services for

positioning systems are typically well balanced for pole sway yet device-specific performance can set a limit for the maximum allowed shaft bending. In this study, car-crash performance was not evaluated but impact-type loads subjected to the pole shaft were deemed an important local failure type.

As depicted in the figure, for both thermal expansion and wind load analyses the poles were assumed to be completely fixed to the ground, i.e., the degrees of freedom were prevented at the base of the pole (0 in every direction), while the loadings were applied over the half surface of the pole outer surface (volume). In detail, the distributed loads were applied on the opposite side of the cutout (in order to test the pole integrity under the highest moment). Respectively, the applied wind load distribution and the temperature difference (thermal load) were equal to 445 N/m² and 20 °C. The nominal element size (edge length) considered in the analysis was 5 mm, and each of the static analyses was roughly constituted by 250,000 elements.

### 2.4. Experimental-Numerical CTE Determination

The experimental setup for determining the CTEs of GFRP consisted of a coupon (projection 252 mm× 60 mm) cut off from the tubular composite pole ($D_{outer} = 168$ mm, see Section 2.1). The coupon was clamped (clamping length 64 mm) to a robust holder from the other end and the free end was subjected to mechanical loading. The mechanical load was subjected by a wire and a free-hanging mass (1.230 kg) attached at a distance of 56 mm from the free end. A strain gauge (KFGS-5-120-C1, Kyowa, Japan) was glued according to the manufacturer's instructions at a 138 mm distance from the clamped end and middle in the transverse direction. The arrangements are illustrated in Figure 5. Finally, the entire setup was placed in a digitally controlled oven. The test included two steps: (1) loading the coupon mechanically; (2) heating the oven in steps (24 °C...60 °C). Each heating step was launched after the strain reading from the gauge had essentially settled.

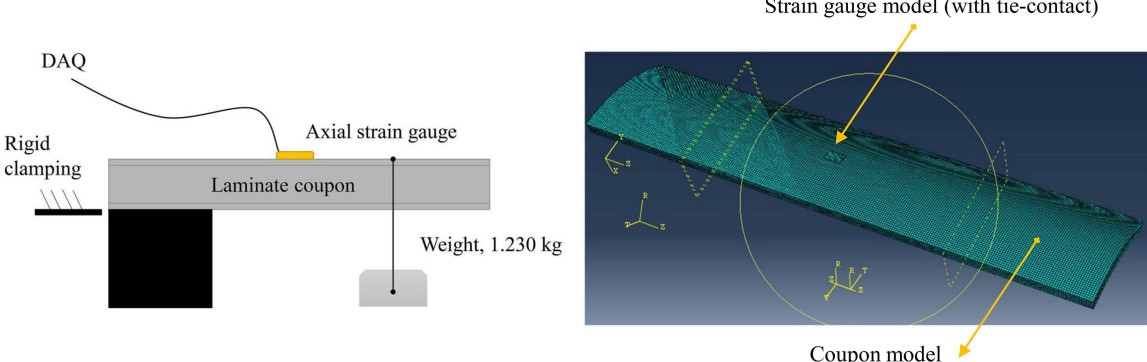

**Figure 5.** Experimental arrangements and the finite element model of the test coupon to determine thermal expansion coefficients.

The finite element model assembly consisted of a coupon model and a gauge model. The composite was modelled using the material constants in Table 1. Because a solid element type (C3D8R) was used, a cylindrical material coordinate system was set so that the radial direction corresponded to the ABAQUS axis nomination '1', the perimeter direction to the nomination '2' and the axial direction to the nomination '3'. The gauge was modelled as a strip of isotropic polyimide (Kapton®) with a Young's modulus value of 4.0 GPa and Poisson's ratio of 0.3 [32]. A CTE value of $1.17 \times 10^{-5}$ 1/°C was used for the gauge based on the adoptable thermal expansion given by the manufacturer (i.e., the gauge follows corresponding expansion in terms of strain reading (zero)). The gauge was meshed by using parabolic (C3D20R) elements. The gauge model was attached to the coupon model by so-called tie-constraints. The model was run by a point load and thermal field ($\Delta T = 36$ °C) in two steps.

The computed strain to match with the experimental strain gauge reading was calculated based on the residual axial stress in the gauge after the thermal load. By presuming that the composite is stiffer than the gauge, i.e., the gauge follows the expansion of the substrate (coupon), we have:

$$\varepsilon_{g,residual} = \sigma_{g,residual} / E_g ,$$ (1)

where $\varepsilon_{g,residual}$ is the residual strain of the gauge in the axial direction, $\sigma_{g,residual}$ is the computed (FEA) residual stress in the axial direction, and $E_g$ is the Young's modulus of the gauge. In Equation (1), it is presumed that the length-change of the gauge is simply due to the 'external' force by the expansion (contraction) of the composite coupon. In reality, the gauge has a finite stiffness and the force balance-given length-change is partly due to the thermal expansion of the gauge (which does not induce stresses). The gauge stresses were recorded from three elements and the average value was calculated.

## 2.5. Signal Attenuation

The sections of 5GPs that provide the needed weather protection to the (5G) radios, are typically referred to as 'radomes'. Low-attenuation and low-permittivity materials are an especial family of polymers that could be employed for the radomes. Radomes mainly protect against moisture and ultraviolet (UV) radiation (e.g., the reference standard UL 746C), as illustrated in Figure 6. The selected material samples' attenuation was measured by using a split-post dielectric resonator (SPDR) at 2.45 GHz (QWED, Warsaw, Poland) with a Microwave Frequency Q-Meter (QWED, Warsaw, Poland). The sample size was 60 mm × 60 mm (thickness 2.5–3.0 mm). The measurements were made at a constant signal frequency of 2.45 GHz and sample-specific thickness was measured to determine attenuation. An especial polymer blend (PREPERM, Premix, Finland) was selected as a candidate material for these details of the pole.

To account for environmental ageing during the anticipated pole operation, a series of samples were conditioned in a UV-chamber. The chamber had UVA-340 fluorescence lamps (Q-Lab, Farnworth, Bolton, UK) with a peak intensity at 340 nm. Each sample set involved five test samples and a set of samples was removed from the chamber and measured at pre-set time intervals. Accelerated aging cycles of 0, 432 h, 864 h and 1728 h (0, 18, 36, 72 days) were analyzed. Due to the durability requirements of sustainable long-term application, black and white paints on the outer surface of the radome materials were surveyed in addition to non-painted samples. The temperature at the chamber varied between 22 °C and 36 °C.

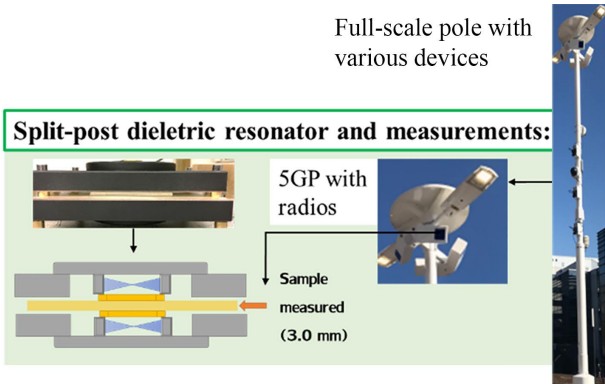

**Figure 6.** Attenuation measurements of radome/pole structure's material samples representing signal windows; radome and measurement setup by using the SPDR method.

## 2.6. Heat Exchangers

The thermal performance of a smart pole is governed by the heat sources, i.e., the devices related to the wireless communication and other services of the product. Therefore, the heat exchanger for

the transfer of thermal energy is a crucial component of the system. Here, aluminum heat exchangers were designed and analyzed for a liquid cooling concept. The motivation of heat exchangers in 5G and other smart pole applications is well-known. The 3D printing was the chosen manufacturing technique because it allows more freedom of design and more complicated shapes. The 3D printing powder was AlSi10Mg and the parts were manufactured by FIT Prototyping GmbH (Lupburg, Germany) by using a 3D printing device SLM 500 (SLM Solutions, Lübeck, Germany), which uses a Powder Bed Fusion technique. The exact geometry of the exchanger is given in Appendix B. The cooling channels were directly 3D printed inside the part to form single integral component.

### 2.7. Computational Fluid Dynamics

The design power range of 100–500 W, as the operating power, required by the 5G radios alone, can be categorized as high-power electronics (HPE). For uniform spatial coverage, multiple directional radios are required leading to ≈1 kW power consumption. The power range is of the same order as the graphics processing units (GPUs) for which liquid cooling systems are commonly employed to maintain moderate temperature levels at components. Hence, one of the ultimate needs of the pole thermal management is the creation of an efficient cooling concept for its HPE. As a common concept for current 5GPs, the initial design was based on air cooling.

For the CFD investigations, the Reynolds Averaged Navier Stokes (RANS) and Large-Eddy Simulations (LES) were utilized [9,33]. In the air cooling simulations, the target was to solve the Navier-Stokes equations along with a transport equation for temperature using the standard, incompressible pimpleFoam solver of the open source CFD code OpenFOAM. The liquid cooling heat transfer simulations were performed with CHT analysis using the standard chtMultiRegionFoam solver in OpenFOAM were the conservation of mass, momentum and energy are calculated simultaneously in both, the liquid and the solid domains. For the heat management, the cooling of four radio units was analyzed. Appropriate cooling capacity requires proper heat exchange and design with optimized internal channeling either inside pole shaft (air cooling) or liquid cooling. Also, a strategy for connecting the cooling medium flow through the four heat exchangers is needed. The 3D printed aluminium heat exchanger was tested for a version family 'V1'. Due to weight, a version family 'V2' and water-cooled heat exchangers were analyzed with the following objectives: (1) minimum material costs, (2) low surface temperatures, (3) more compact size than V1 versions, and (4) the system should be functional even if lukewarm water is available. The simulation parameters are given in Table 2. The design concept with flow and heat values are given in Figure 7. The model consisted of 11 million cells in the fluid domain and three million cells in the solid domain—hexagonal cells except at the edges/interfaces cubic cells were used.

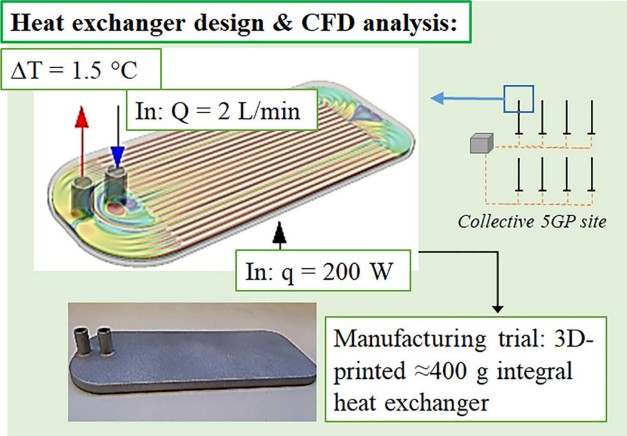

**Figure 7.** The cooling design for an integral device configuration in the smart pole(s): the estimated input data for thermal analysis of a heat exchanger by using CFD.

**Table 2.** Input selections used for the CFD analysis of heat management with 3D printed heat exchangers.

| Fluid Domain | Solid Domain |
|---|---|
| Turbulence model used: $k$-$\omega$ SST | |
| **Inlet:** | **Heated surface:** |
| Velocity 0.4 m/s | Fixed temperature gradient |
| Turbulent intensity 4.00% | Zero pressure gradient |
| $k = 0.00038$ m$^2$/s$^2$, $\omega = 51.57$ 1/s | **Fluid-solid walls:** |
| Temperature 300 K | Temperature calculated |
| Pressure with zero gradient | Zero pressure gradient |
| **Walls:** | **Outer surfaces:** |
| Velocity—no slip | Zero temperature gradient (adiabatic) |
| Turbulent variables—wall functions | Zero pressure gradient |
| Zero pressure gradient | |
| Temperature calculated | |
| **Outlet:** | |
| Zero velocity gradient | |
| Zero gradient of turbulent variables | |
| Fixed pressure (atm pressure) | |
| Zero temperature gradient | |

## 3. Results

### 3.1. Impact Dynamics of the Selected GFRP Shaft

During the design process, prior to manufacture of full-scale products, experimental validation and qualification was started in phases. For a composite 5GP, even for no-traffic sites, the critical damage is impact-type loads at the shaft root. In this study, an experimental campaign was realized by a step-by-step approach starting from QS indentations on GFRP panel sections and, further, to full-scale impact tests on the tubular shaft (more details in Section 2.2). The QS testing serves as a limit case (reference and control) for impact-concerned design since dynamic effects are omitted. In general, the deformations of the curved specimens clearly localized close to the contact areas. The localization for the GFRP panel can be seen in Figure 8, where the deformations of the lower surface in the indentation case (at the maximum loading moment) are evident. The localization challenges the sizing process of the pole since the ultimate (fracture) behaviour starts playing a big role in the GFRP's deformation response.

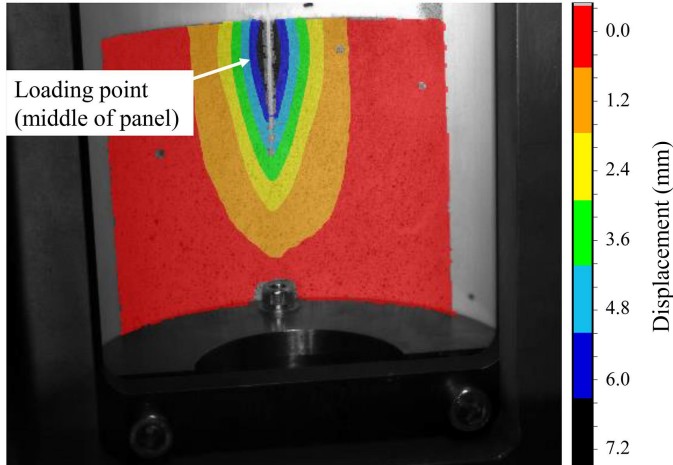

**Figure 8.** The mechanical testing of a panel specimen: the DIC-resolved displacement field during the indentation loading of the half-circular specimen at the moment of maximum load.

The contact force versus displacement of the loading head was defined for different loading configurations during the testing campaign (see Figure 9). The results show clearly the effects of specimen size and support on the load response. For the similar size half-circular specimen, the trend of the loading in the QS indentation and drop-weigh impact is essentially similar regardless of the difference in the support (semi-rigid or open boundary). However, when comparing the impact response of the tubular specimen to the half-circular panel, the indentation and impact cases showed clear differences in terms of maximum load and ultimate deflection. Figure 9 also includes local FEA of the GFRP panel (failure criteria applied) [27]. It was confirmed, as is typical for composites, that the load-carrying capability after the damage onset remains. The first failure mode due to the impact was internal delamination and often visible crack in the axial direction of the shaft.

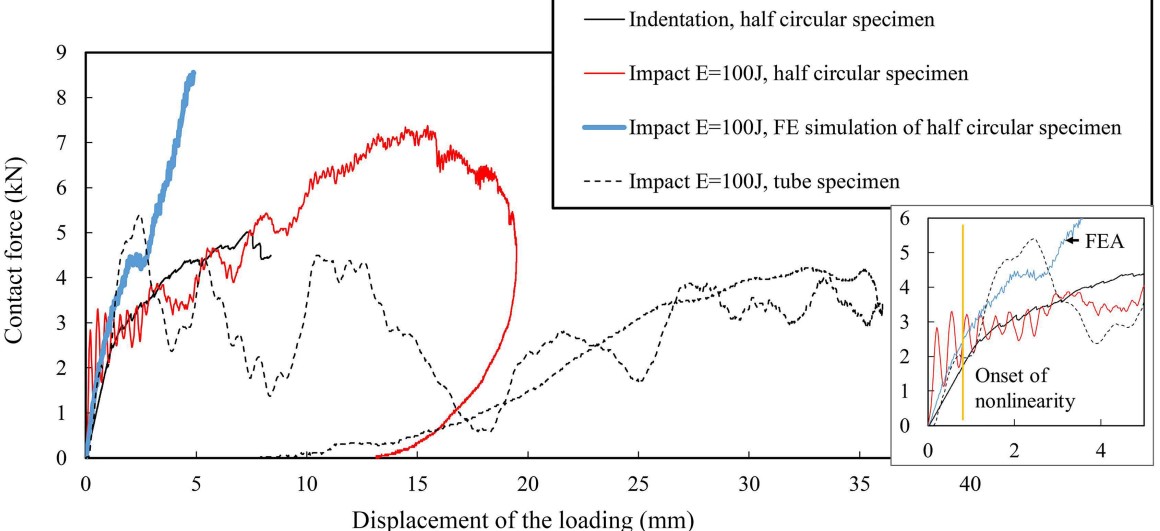

**Figure 9.** The performance of the GFRP laminate and tube: the contact force-displacement response based on experiments and FEA (linear material [27]). The inset graph on the right shows the detailed data at the beginning of the tests/analyses and predicted displacement level for damage onset.

### 3.2. Finite Element Analysis

FEA was used to predict the effects of cutouts (maintenance doors) when the pole is subjected to the wind load and thermal load (e.g., due to radiation from the sun). For most of the design cases, the critical details are the cutouts designed for connections (cabling in and out from the pole) or maintenance, typically located at the shaft root with a high level of bending moment. Figure 10 shows the failure analysis for the diamond-shaped cross-section at the root cutout, where the maximum stress criterion predicts the onset of damage. The exact type and form of the failure criterion required analysis for the curved shape and different strain-rates [34]. Finally, the confirmation of the selection was made by comparing with the experiments (see Section 3.1). A 5 mm-wall (for the load-carrying layers) thickness represented a safe and low-weight solution.

For the composite shaft, the thermal strains (per material's coefficients of thermal expansion) are governing the absolute deformation of the composite pole for the anticipated, average operation environment, i.e., non-storm weather. For this reason, additional thermal analyses were carried out together with the wind load analysis (Figure 11). The pole design with the circular cross-section experienced 15–64% higher deformations (51 mm compared to 84 mm with wind load and 64 mm compared to 73 mm with thermal load for the diamond-shaped and circular cross-section, respectively). Due to the anticipated services of the project (see Section 4.2), the level of deformations was not seen problematic and the lower costs of the circular cross-section overran the mechanical advantages of the diamond-shaped cross-section.

**Shaft stress analysis:**
Failure index (maximum stress criterion)

Wall thickness 5.0 mm　　　　　　　　　Wall thickness 8.0 mm

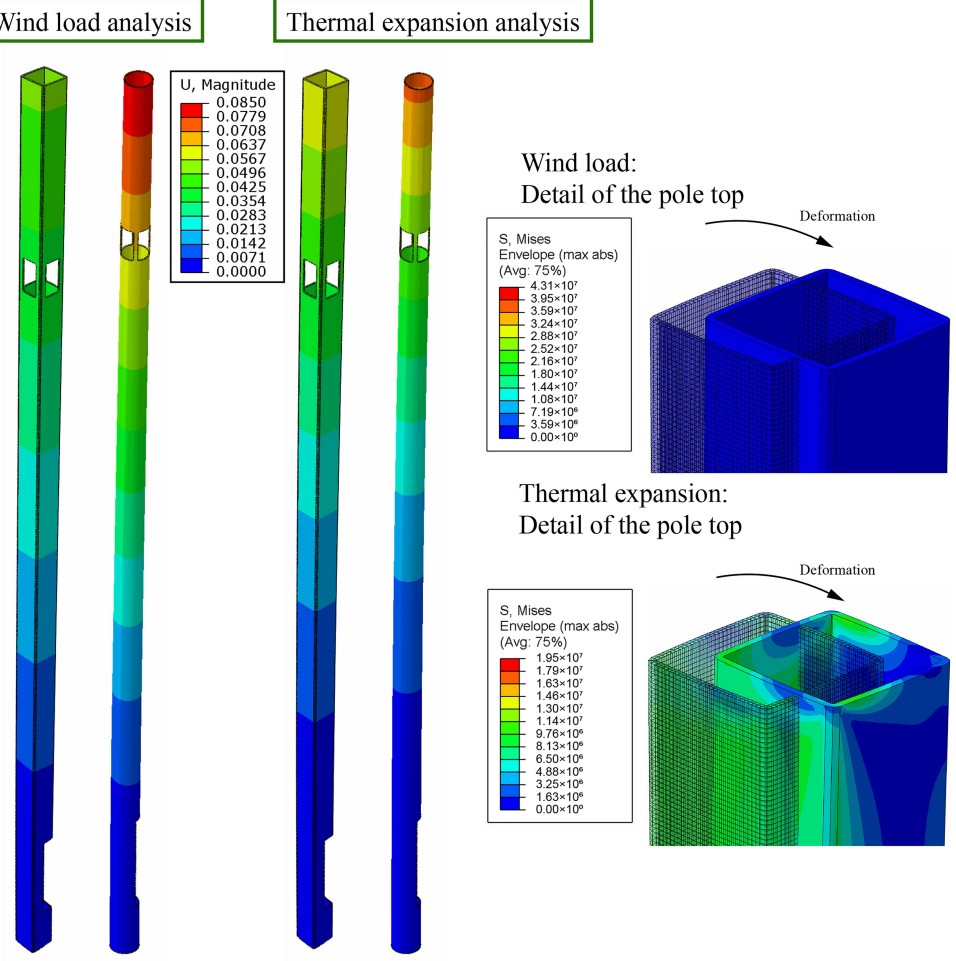

**Figure 10.** FEA for the failure index when comparing two different pole shaft laminates and a diamond-shaped cross-section (5 mm and 8 mm-wall thickness); corners (hotspot) at a root cutout are shown.

**Figure 11.** FEA for the deformation when the pole shaft is subjected to the standard wind load (**left side image**) and a thermal load (**right side image**). Details of the highest deformation (pole top) are shown on the right.

### 3.3. Signal Attenuation at the GHz-Regime

Any enclosing of the (5G) radios inside a housing or shaft requires analyzing the RF signal attenuation due to the surrounding enclosure or radome wall. The design of the details, i.e., radomes or 'signal windows' in cutouts, must satisfy experimental verification regarding the attenuation. In this study, the radomes were designed not be load-carrying parts so that reinforcements (fibers) were not needed for the radomes. PREPERM polymer (see Section 2.5) was analyzed here, and attenuation defined in terms of dielectric loss (DL). In particular, the long-term properties of the radome materials in outdoor environments were not well-known. Therefore, the effects of UV radiation (from the sun) were considered in this study. In reality, the signal windows might need to be painted that makes the measurements with painted samples necessary. According to the measurement results, shown in Figure 12, the minimum paint design is crucial for a high signal penetration. The effect of a paint layer had a significant impact on the signal attenuation (i.e., dielectric loss (DL)) and the treatment increased the DL levels 90–175%. Slightly lower increase was measured as the UV degradation was increased (longer UV exposure time); the color of the paint did not observably affect the attenuation.

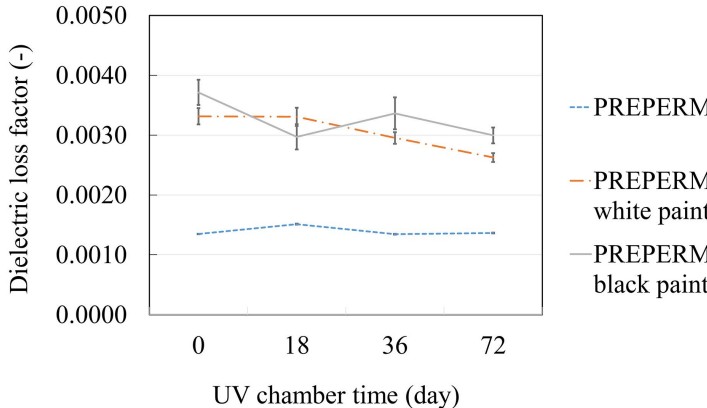

**Figure 12.** Measured attenuation in terms of DL; results over the 72-day accelerated UV-ageing period.

### 3.4. Thermal Management and Multiple Radio Analysis

The CFD analysis process was divided into: (1) CFD investigations, (2) feasibility study of 3D printed heat exchangers, and (3) experiments on a 3D printed heat exchanger operating at 200 W. The air-cooling systems by standard plane and pin-fin designs led to heat exchangers with a weight of 3.5–3.8 kg per piece and their volume was overly large compared to the space available inside the demo 5GPs. With room temperature air, the system's maximum surface temperatures remained below 51 °C that is substantially below the 65 °C critical allowable tempareture. Concerning the CFD simulations, the constant surface temperature-boundary condition was noted to be misleading as: (1) the surface temperature is not known, and (2) the inserted power cannot be fixed. In the experimental validation, the simulated and experimental velocity fields (using Laser Doppler Anemometry) agreed well. Conjugate heat transfer (CHT) studies are proposed as the next design steps of the design so that temperature transport in both fluid and the solid can be accounted for.

One of the key drivers for a cooling concept is the compact size and centralized thermal control at an entire 5GP site. A liquid cooling was a tempting option to be analyzed since the density ratio of water to air is 1000:1 while the specific heat ratio is 4:1. Furthermore, water is much better heat conductor than air with the heat conductivity ratios of approximately 60:1. Motivated by the experiences from the air cooling investigations, we utilized the CHT solver in OpenFOAM called chtMultiRegionFoam. With this procedure, the incompressible Navier-Stokes equations were solved for the fluid phase while the convection-diffusion equation was solved for temperature both in the solid and fluid.

Based on the results, the walls could be made thinner (e.g., 1 mm) to result in a less than 200 g mass of the exchanger while the tested exchanger version (wall thickness 2 mm) had a ≈393 g mass. It was confirmed by simulations that the system can maintain surface temperatures below 65 °C for a range of mass flow rates for cooling water's inflow temperature below 50 °C (see Figure 13). Figure 13b shows the inlet temperature of the cooling water as a function of the mass flow rate that maintains the heat exchangers' heated surface temperatures at 65 °C. The serial and parallel configurations indicate whether the heat exchangers (inside) of the 5GP would be connected in series or in parallel to cool all the four radios. The highest allowed inlet water temperature for the serial and parallel configurations of liquid cooling heat exchanger is based on the assumption that the temperature difference between the inlet water and the heated surface is independent of the absolute temperature—based on the mass flow rate sensitivity analysis that has confirmed the independency of transfer performance on the inlet Reynolds number [33]. Furthermore, the functionality of the heat exchanger was tested experimentally, and the measured surface temperatures were in a good agreement with the CFD results.

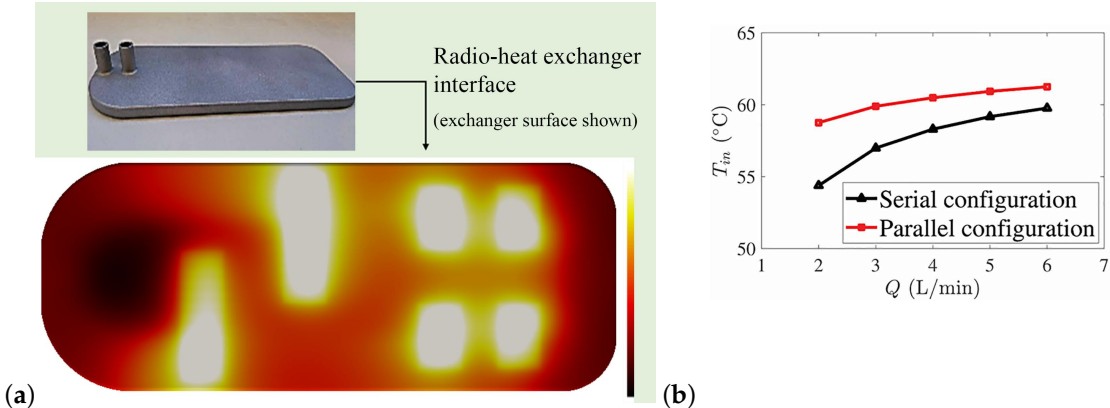

**Figure 13.** CFD analysis of a 3D printed heat exchanger and the cooling strategy: (**a**) simulated heat exchanger (outside) surface temperature (color range represents $\Delta T = 10$ °C) when water flows into and out from a 3D printed heat exchanger via embedded rows of channels; (**b**) the highest allowed inlet temperature as a function of mass flow rate to maintain below 65 °C surface temperatures per cooling strategy.

## 4. Discussion

### 4.1. Design Process and Interconnections of Results

The FEA combined with the experimental campaign resulted in the GFRP wall thickness of 5 mm (load-carrying layers). For a six-meter pole and GFRP's density of 1670 kg/m³, the mass of the entire pole would be 28.8–29.6 kg (from circular to diamond-shaped cross-section, respectively). For a similar steel pole, the mass would increase 380% (steel pole mass 153 kg). Because 5GPs are more deformation (sway) critical than strength critical, a steel pole could be made six times thinner in thickness in theory. If a practical minimum wall thickness would be two millimeters to prevent instability, the steel pole continue to be 60% heavier (i.e., GFRP leads to a minimum of 37% weight saving). The significantly lower mass of the GFRP pole directly makes its handling easier and lowers transportation emissions for a global 5GP usage.

Because of the bending moment and the following stress concentrations at the root of the 5GP shaft, the wall thickness could be increased at the pole root. Alternatively, the pole root in turn would be an ideal location for an additional, fully or partly load-carrying structure, i.e., a wide shaft tube. A wide housing at the root would lower the mechanical stresses and embody a neat space for standard power and data connection modules. A root housing could also accommodate devices, e.g., an EV charge station for some urban 5GP sites. Due to the linearity and strength of composite materials,

there are clearly more options, e.g., large maintenance doors, as was given by the FEA at the cutout corners in this study. At the top of the pole, cutouts are also necessary for radio radomes. Large cutouts for signal windows (covered by a non-reinforced polymer) make the GFRP pole even lighter. Any other than composite/polymer design could be an obstacle for receiving (indoor/outood) devices [26] and to fit the 5GP for individual customer needs. The attenuation measurements indicated a 90–175% increase of signal attenuation due to a surface treatment on radome materials—this means that as large cutouts and as thin as possible radomes are needed in 5GPs, even when using GFRP for the pole shaft.

The CFD analysis of the heat management presented that liquid cooling is an efficient technology for pole-integrated radios. Liquid cooling requires piping between individual radios as well as in and out from the pole. The added pipe lines would require further space inside the pole; the power and data cabling with various connector appliances define the necessary pole diameter in general. Any larger diameter will lead to a heavier pole—again emphasizing the benefits of GFRP.

The results of this study showed that the interaction between functionality (i.e., large cutouts allowing large radomes and maintenance doors), heat management for integrated devices and mechanical design with a minimum material usage and safe structure prefer a composite design. Due to the higher manufacturing costs of composite structures compared to traditional steels, the markets of 5GPs will define the amount of device integration in the future.

### 4.2. Assembly and Future Applications

Currently, smart poles as commercial products are complex systems with various stakeholders involved. The division between a product owner, seller, data handling, etc., has not yet settled. As a European solution, the consortiums of Luxturrim5G, Neutral Host and Luxturrim5G+ [35] ventures have defined a 5GP overall concept that must deal and handle all the issues of future wireless platforms: legislation, radios, big data handling for 'Smart cities', viable business concepts, the open data platform and safe physical integration at urban areas. The full-scale 5GPs were mounted at the Karaportti site (Espoo, Finland) in the autumn 2019. The 5GPs of the site finally included the following services: 60 GHz WiGig radios, video and audio surveillance, weather monitoring, and EV charging. Part of the devices and all the power and data cabling were pole shaft integrated. The connection with the city infrastructure was analyzed using a 3D planning tool (AURA, Sitowise Oy, Espoo, Finland) prior to the excavation work. In 2020, activities continue by mounting the future 5GPs with 60 GHz WiGig and 26 GHz radios, as well as new services for traffic monitoring, autonomous driving and public safety.

### 5. Conclusions

Several potential designs of 5GPs have been proposed for the physical device frame and service platforms within the current industry of wireless communication technologies. This study focuses on the physical structure of a 5G smart light pole and its multidisciplinary design process. The work includes an interacting research of a GFRP composite pole structure with finite element (FE) analysis and experimental verification, signal attenuation measurements of the latest low-attenuation materials and metal 3D printing combined with high-fidelity CFD computation to understand the heat management inside the densely device-integrated 5G pole. Based on the results, the work revealed the following specific novelties related to the next-era wireless application platform:

- A full-composite glass fibre reinforced 5G pole was FE-modelled and analysed against standard wind and thermal load. The findings showed that a mechanically safe and functional (stiff) GFRP shaft results in significant weight savings (37–80%) compared to traditional steel shafts;
- RF signal attenuation at a GHz-regime (2.45 GHz) was found to increase significantly (90–175%) due to any paint layer while long-term UV degradation in the polymer structure led only to a nominal decrease of attenuation in terms of dielectric loss;
- Entirely integral one-piece heat exchangers were designed with CFD analysis of the fluid-solid interaction for heat transfer, and printed. It was found that a parallel liquid cooling of four

radio units is rather insensitive to the flow rate (range 2...6 L/min) and as high as ≈60 °C inlet temperature can be allowed to keep the device surfaces at or below a critical 65 °C.

**Author Contributions:** Conceptualization, M.K. and V.V.; methodology, T.P. and K.S. and H.H.; software, D.D.V. and A.L. and M.K.; validation, D.D.V., A.L. and T.P.; investigation, J.J. and K.K.; writing—original draft preparation, M.K. and D.D.V. and T.P.; writing—review and editing, V.V. and H.H.; visualization, M.K.; project administration, M.K. and H.H. and V.V. All authors have read and agreed to the published version of the manuscript.

**Funding:** This work was done as part of the LuxTurrim5G ecosystem funded by the participating companies and Business Finland.

**Acknowledgments:** Researchers J. Jokinen and O. Orell are acknowledged for their help and support. Nokia Bell Labs (J. Salmelin and P. Wainio) are acknowledged for the collaboration and design process development. Exel Composites (M. Lassila and K. Sjödahl) is acknowledged for the material and manufacture support.

**Conflicts of Interest:** The authors declare no conflict of interest. The funding agency had no role in the interpretation of data.

## Appendix A. Thermal Expansion of the GFRP Composite

The results of the CTE determination for GFRP are shown in Figure A1. The error between the experiment and FEA only for the mechanical load was 13% (FE model stiffer, $\varepsilon_g = 4.18 \times 10^{-5}$). For the combination of mechanical load and thermal load, the target deviation (error) was kept the same. The experimental strain developed slowly per temperature step, and an extrapolated value of $\varepsilon_{g,residual} = -1.6 \times 10^{-4}$ was selected as the target. Finally, an iterated solution was met by using the values of $CTE_{axial} = 1.15 \times 10^{-5}$ 1/°C and $CTE_{transverse} = 2.5 \times 10^{-5}$ 1/°C ('axial' corresponding to the '3' direction and 'transverse' corresponding to the '2' and '1' directions in the FEA). This iteration was deemed acceptable due to the fact that there are no non-linearities accounted for in the FEA that in reality might appear (e.g., at gauge-coupon interface or in the gauge). Additionally, Equation (1) does not take into account the deformation of the substrate (coupon) by the expansion of the gauge. These effects would lower the calculated strain with the selected CTE values [36]. It should be noted that the transverse CTE of GFRP affects the gauge strain (Equation (1)) due to the Poisson's effect (in FEA). The iterated CTE values are in a good agreement with the literature values for GFRP [37,38].

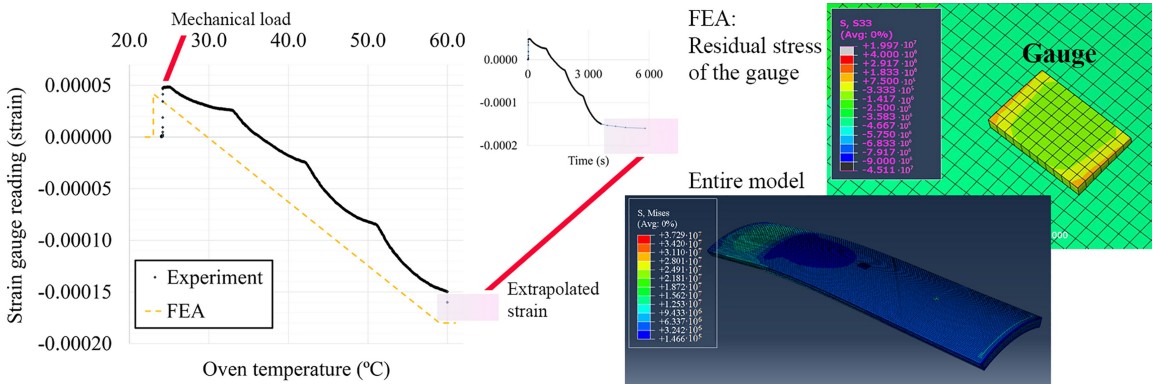

**Figure A1.** The experimental and the finite element analysis results used for the determining of GFRP's thermal expansion coefficients.

## Appendix B. Heat Exchanger Design

The dimensions of the 3D printed integral heat exchanger are given in Figure A2.

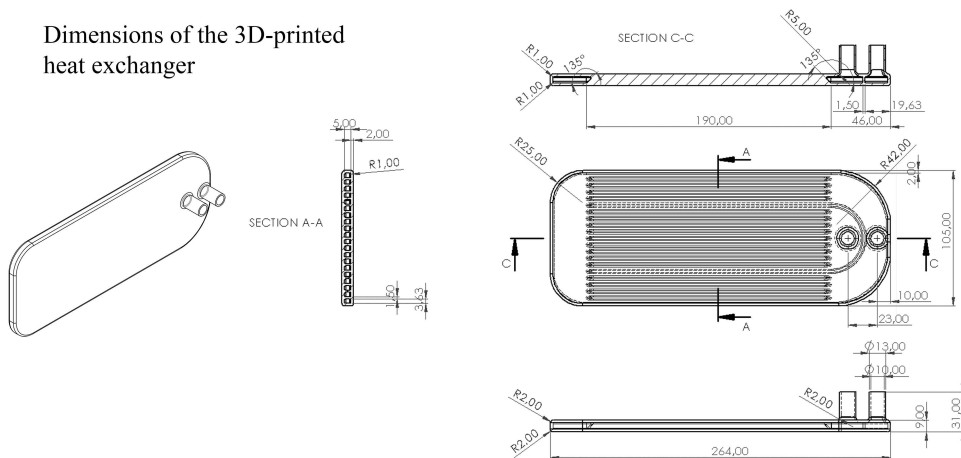

**Figure A2.** Technical drawing of the heat exchanger. Dimensions are in millimeters.

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
