# Peer review of "Safe and Sustainable Design of Composite Smart Poles for Wireless Technologies"

_applsci, doi:10.3390/app10217594_

Round 1
Reviewer 1 Report
This is a very interesting article, relevant to the current discussion about 5G. I recommend it for publishing in current version.
Reviewer 2 Report
The topic of manuscript is interesting to many readers, and it can be seen from the whole article that the authors have done a plenty of efforts on this work. The writting looks excelent. However the organisation/structure of the article seems a bit confused. For instant, the resaerch method is presented behind the results/discussion, the experimental setup of CTE measurement is presented as an appendix.
The proposed methodology for the design of the pole is fine. The authors have tried to cover many aspects that relate to the design of a GFRP composite pole, however the overall content seem very diverted. I can hardly see any innovation through these individual test/modelling, and the conclusions did not show the significant of contribution.
Reviewer 3 Report
Thanks for the manuscript.
The article aims to study safe and sustainable design of composite smart poles for wireless technologies focusing on the physical structure of a 5G smart light pole and its multidisciplinary design process.
I appreciate the work of the authors, ... however, the presentation of the investigation is very bad organized and the line of the research is not clear. To to be the article worth publishing, it needs complete reorganization at which the scientific approach and the individual stages of research (analysis, data synthesis, results´ processing and evaluation, discussion and conclusions) should be clearly manifested. The authors should take into account the following comments at the article improvements:
- Introduction and state of the art (so, e.g. sections like 3.1. The advantages ... and 2.1. should be incorporated here - within Introduction).
- How can the Chapter "Results" be included directly after "Introduction", if the basic conditions of the research are not known? If ...
- the testing methodology is not clear,
- a detailed description and characteristics of the test subjects are not given,
- dimensions (overall or shape elements) are not given,
- material specifications, composition and physical or mechanical properties are not relevant for the evaluation of the results obtained, from which the test subjects are or will be made
- there is no description of the methodology for evaluating the results
- Figures 4 and 5 present the results of the numerical analysis, but no boundary conditions of the analyzes are known (load, bonds, temperatures, coefficients and other characteristics for starting and correct evaluation of FEM analysis including the type of element of the finite element network, number of elements, their distribution within the network, contact conditions ...)
- Section „3. Discussion“ doesn´t provide any analytical discussions and evaluation related to the achieved results.
- Several objects have been tested ... but it is not clear what is a connection between the objects, whether tests and analyzes should follow each other, whether the results of individual components are mutually conditioned.
- Conclusions have to be more specific and based on the obtained results.
Round 2
Reviewer 3 Report
Thanks for the improved manuscript!
I studied the rewritten article carefully and I need to say that the authors did a good job. In my opinion, it is suitable for publication now as it is.